# Professional identity and its relationships with AI readiness and interprofessional collaboration

**Wafa'a Ta'an**[1]*, **Sadeq Damrah**[2], **Mohammed M. Al-Hammouri**[1], **Brett Williams**[3]

**1** Community and Mental Health Nursing Department, Faculty of Nursing, Jordan University of Science and Technology, Irbid, Jordan, **2** Department of Mathematics and Physics, College of Engineering, Australian University, Safat, Kuwait, **3** Department of Paramedicine, Monash University, Clayton, Victoria, Australia

* wftaan@just.edu.jo

## Abstract

### Background

In contemporary healthcare practices, the convergence of Artificial Intelligence (AI) and interprofessional collaboration represents a transformative era marked by unprecedented opportunities and challenges. The introduction of AI technologies is assumed to lead to changes in the nature of interprofessional collaboration that require revisiting the already established professional identity; however, research is lacking in the area.

### Objective

To examine professional identity and its relationships with AI readiness domains and interprofessional collaboration components.

### Methods

A multisite cross-sectional research design was used to recruit 512 participants from different healthcare professions in Jordan between November 14th, 2023, and February 13th, 2024. The Medical Artificial Intelligence Readiness Scale and the Readiness for Interprofessional Learning Scale were used in data collection. Data analysis included descriptive, correlation, and comparative analyses.

### Results

Professional identity significantly and positively correlated with artificial intelligence readiness total and subscale scores with ρ ranging from 0.37 to 0.47 (p<.01). In addition, professional identity significantly correlated with interprofessional teamwork and collaboration (ρ=0.79, p<.01) and the roles and responsibilities components of interprofessional collaboration (ρ=0.37, p<.01). Professional identity was significantly higher among male participants and participants with experience of five years or higher.

**Data availability statement:** The datasets generated and analyzed during the current study are available as a supplementary file accompanying this manuscript.

**Funding:** This study is funded by the Deanship of Research at Jordan University of Science and Technology (Research Grant ID: 20230635). The funders had no role in study design, data collection and analysis, decision to publish, or preparation of the manuscript.

**Competing interests:** The authors have declared that no competing interests exist.

## Conclusion

The study sets the grounding roles to develop the healthcare workforce's professional identity within the dynamic healthcare environment in the age of artificial intelligence and interprofessional collaboration. The study highlights areas of development for healthcare managers and practitioners, such as AI interprofessional collaboration-based training, targeting both artificial intelligence domains and interprofessional collaboration components while preserving a positive professional identity.

## Introduction and background

In contemporary healthcare practices, Artificial Intelligence (AI) represents a transformative era marked by unprecedented opportunities and challenges [1]. AI technologies continue to expand widely into all aspects of a human's life including education, research, and healthcare [2]. AI is defined as the ability of a machine, computer, or technology to perform tasks that require the intelligence of a human being [3]. Applications of AI in the healthcare field include but are not limited to areas like diagnosis and prediction, robotic surgeries, medical imaging, and drug discovery [4]. A literature review was conducted to track the scientific accomplishments of AI in biomedicine indicating that the application of AI is still at an early stage that requires breakthroughs and horizontal and vertical improvement to improve deepen and expand AI utilization and advancement in healthcare [5]. This requires an investigation of healthcare professionals' readiness for AI utilization.

AI readiness in healthcare refers to the capacity of healthcare professionals and organizations to effectively utilize, integrate, and adapt to AI technologies [6]. This concept encompasses four dimensions; namely, cognitive, ability, vision, and ethics [7]. However, AI integration requires interprofessional collaboration (IPC) in healthcare highlighting its transformative potential to influence teamwork, communication, and decision-making [1]. The readiness of healthcare professionals to embrace and integrate AI technologies is a critical determinant of the industry's ability to leverage the benefits of AI advancements [8]. Simultaneously, the principles and practices of IPC play a pivotal role in shaping the professional identity of healthcare practitioners, especially within this evolving technological environment.

Although technological advances such as AI have great potential, contradictions and uncertainties surrounding professional identities exist [9]. A qualitative study was recently conducted to uncover the influence of emerging technologies on professionals' future roles and identities using a thematic analysis. The main themes were embracing AI to improve efficiency and continuous professional development, perceiving anxiety about the potential replacement of professionals by artificial intelligence and robots, and recognizing that the human essence is irreplaceable in terms of critical thinking and empathy [10].

While there has been limited exploration of the interplay between professional identity and attitudes toward the adoption of AI, existing literature suggests some

indications of the potential influence of professional identity on the adoption of novel technologies. For example, pedagogies, goal orientation, and approaches/transformation to service delivery may be affected [9]. Furthermore, consideration and understanding of the dynamics of professional identity within the realm of IPC is essential for fostering a collaborative environment that maximizes the collective potential of healthcare professionals [11]. For example, classic work on IPC used to include health-based professionals only, however, with AI integration an extension must be considered to include professionals in the computer and technology departments. Therefore, the introduction of AI technologies is assumed to lead to changes in the nature of IPC that require revisiting the already established professional identity.

Therefore, the professional identity pedagogies and roles and responsibilities among healthcare professionals such as nurses and physicians may need to be revisited to adapt to the advances of AI. Professional identity can be envisioned as a continuum that extends from negative to positive poles. This spectrum captures the evolving nature of how individuals perceive and define themselves within their professional roles. On the negative end, challenges or conflicts may shape one's professional identity adversely such as oppression and poor professional image [12,13] which can lead to negative consequences such as turnover [14]. On the other hand, positive professional identity is associated with the professional's power, autonomy, and enhanced self and group esteem [15]. Positive professional identity is linked to increased motivation and enhanced care quality which is the ultimate goal for all healthcare institutions [14].

Professional identity develops as a consequence of the collective interaction between individuals and surrounding circumstances within a given context [16]. As professionals engage with AI technologies, their roles may evolve, necessitating a reevaluation of professional identity constructs [17]. Positive aspects, such as adaptability and a commitment to continuous learning, can be fostered through effective IPC, contributing to a more resilient and dynamic professional identity [18,19].

Professional identity is considered an integral aspect of IPC; however, teamwork, collaboration, and roles and responsibilities within interprofessional practice are also fundamental components of IPC [11,20]. The collaborative attitude promoted by IPC can foster a climate of shared responsibility, mutual respect, and effective communication for navigating the dynamic complex landscape of healthcare. IPC emphasizes the collective effort of professionals from various educational backgrounds working collaboratively to optimize patient care outcomes [21,22].

Donovan and colleagues stressed that IPC requires collaboration among a team of healthcare professionals with overlapping proficiency while respecting the unique contribution of each team member toward achieving a common goal [23]. Although unraveling the construction of professional identity will undoubtedly contribute to the cultivation of a collaborative ethos, and ultimately enhance the quality of patient care, limited research has addressed professional identity for professionals in healthcare settings [11,24]. For example, the professional identity of healthcare practitioners may be shaped by the intersection of AI readiness and IPC.

To unpack the complexity of professional identity construction amidst modern healthcare, this research aims to investigate the role of AI readiness, and IPC components in evolving professional identity of healthcare practitioners. Understanding how AI readiness and IPC intersect to shape professional identity is vital for healthcare professionals, educators, and policymakers as they navigate the evolving landscape of healthcare delivery. The specific objectives of the study are:

1- To describe the sociodemographic composition of healthcare practitioners.

2- To assess the scores of AI readiness and its domains (cognitive, ability, vision, and ethics), and IPC components (professional identity, teamwork and collaboration, and roles and responsibilities).

3- To examine the relationships between professional identity and AI readiness domains and IPC components.

4- To assess the differences in professional identity scores according to the participants' sociodemographic characteristics.

 

## Methods

### Study design and sampling

This study utilized a multisite cross-sectional correlational comparative design. A convenience sampling technique was used to collect data from professionals working in three hospitals in Jordan. The inclusion criteria were being a professional working in the hospital, having a full-time appointment, and having a minimum experience of one year. The sample size was calculated using the G*power analysis software with an effect size of 0.25, alpha of 0.05, power of 0.9, ten covariates, and an additional 15% was added for nonparametric analyses. The calculation resulted in a minimum of 460. The final sample size was 512 participants.

### Measures

The study kit included a sociodemographic sheet with questions about the profession, educational level, experience, and gender, followed by two questionnaires. The first was the Medical Artificial Intelligence Readiness Scale (MAIRS) which is a 22-item four-dimensional scale [7]. The questionnaire measures AI readiness on the cognitive, ability, ethics, and vision dimensions. The participants can rate their responses on a 5-item Likert scale ranging from strongly disagree (1) to strongly agree (5). This questionnaire was previously tested and proved valid and reliable with Cronbach's alpha values for the subscales of cognitive, ability, vision, and ethics domains as.93,.95,.91, and.31, respectively, and for the total scale was.97 representing excellent reliability [7].

To measure professional identity and other IPC components, the Readiness for Interprofessional Learning Scale (RIPLS) was used [20]. This questionnaire includes 19 items and has three components which are teamwork and collaboration (T&C), professional identity (PI), and roles and responsibilities (R&R). The responses can be recorded on a Likert scale ranging from strongly disagree (1) to strongly agree (5). The RIPLS was assessed for psychometrics and found valid and reliable with a total score's Cronbach alpha value of 0.89. and 0.88, 0.81, and 0.43 for T&C, PI, and R&R components, respectively [20]. To allow for subscale comparisons for both measures, the standardized mean score was calculated by dividing the mean scale value by its corresponding number of items.

Both MAIRS and RIPLS are commonly used and were designed to assess readiness in artificial intelligence in healthcare and interprofessional learning, respectively. Their alignment with the study's goals made them appropriate choices, even with some limitations in internal consistency. To address these challenges, the reliability scores of these measures were assessed among the study sample using Cronbach alpha and item-total correlations analyses. The results demonstrated good to excellent readings, see the results section.

### Ethical considerations

The study proposal was granted ethical approval by the institutional review board of Jordan University of Science and Technology (1/165/2023). Potential participants were informed about the study's purpose and assured that their responses' confidentiality and privacy would be maintained. The participants were also informed that participation is entirely voluntary before consenting to participation. A written informed consent was obtained from each participant before completing the study questionnaire.

### Data collection procedure

Data collection occurred between November 14th, 2023, and February 13th, 2024. The administrators at the data collection sites were contacted to obtain permission and facilitate access to potential participants. Three trained research assistants handled the data collection, one in each hospital. They approached potential participants, provided information on the study, screened them for eligibility, distributed the questionnaires, collected them back after completion, and checked that they had consented to participation. The research assistants were firmly asked to include hospital personnel from different

professions as long as they meet the eligibility criteria because the study's concepts are not limited to certain groups of professionals within the healthcare industry.

### Response rate and missing data handling

To account for incomplete questionnaires an additional 20% was added. So, 552 questionnaires were distributed, among which 519 were returned, and seven of them were excluded because they had missing responses for more than a third of the questionnaire. However, when the returned questionnaires were found missing a few values at random, the missing values were replaced by the item's mean value. The final sample included in the analysis was 512 participants with a response rate of 92.7.

### Data analysis

The Statistical Package of Social Sciences version 27 [25] was used in this study. The data were entered, cleaned, screened, and preliminarily checked. Participants' characteristics and main study variables were analyzed using descriptive statistics, including frequencies, percentages, mean, and standard deviation. The scales and subscales' internal reliabilities were assessed using Cronbach's alpha.

The preliminary analysis revealed that the professional identity scores were not normally distributed among the participants nor among the groups of participants which require nonparametric comparative analyses. Therefore, the Mann-Whitney U test was used to compare professional identity scores among bicategorical groups and the Kruskal-Wallis test was used to compare professional identity among the participants' groups of professions. Additionally, Spearman's rank correlation coefficient was utilized to examine the relationships between professional identity and other variables (AI readiness and IPC components).

## Results

### Participants' characteristics descriptives

The study sample was composed of 512 participants including nurses, physicians, midwives, pharmacists, allied medical personnel, computer and information technology specialists, and others (e.g., lab technicians & health record professionals). The gender distribution of the participants was 37.9% for males and 62.1% for females. The majority of the participants held a Diploma or BSc degree (92%), and only 8% obtained post-graduate degrees (8%). Regarding experience, about half of the participants (56.8%) reported less than 5 years, while 43.2% reported 5 years or more. Table 1 presents a detailed analysis of the sociodemographic characteristics of the participants, stratified by job title. The distribution of gender, educational levels, and professional experience is summarized for each professional group.

**Table 1. Sociodemographic characteristics of participants according to job title (N = 512).**

| Job title | Gender | | Educational level | | Experience | | |
|---|---|---|---|---|---|---|---|
| | Males | Females | Diploma or BSc | Post-graduate | <5 years | ≥5 years | Total (%) |
| Nurse | 69 (45.3) | 116 (54.7) | 175 (82.5) | 37 (17.5) | 91 (42.9) | 121 (57.1) | 212 (41.4) |
| Physician | 58 (48.3) | 62 (51.7) | 120 (100) | 0 (0) | 100 (83.3) | 20 (16.7) | 120 (23.4) |
| Midwife | 0 (0) | 82 (100) | 82 (100) | 0 (0) | 58 (70.7) | 24 (29.3) | 82 (16) |
| Pharmacist | 8 (28.6) | 20 (71.4) | 28 (100) | 0 (0) | 16 (57.1) | 12 (42.9) | 28 (5.5) |
| Allied Medical Personnel | 12 (46.2) | 14 (53.8) | 22 (84.6) | 4 (15.4) | 10 (38.5) | 16 (61.5) | 26 (5.1) |
| Computer & IT Specialist | 12 (75) | 4 (25) | 16 (100) | 0 (0) | 4 (25) | 12 (75) | 16 (3.1) |
| Others | 8 (28.6) | 20 (71.4) | 28 (100) | 0 (0) | 12 (42.9) | 16 (57.1) | 28 (5.5) |
| Total | 194 (37.9) | 318 (62.1) | 471 (92) | 41 (8) | 291 (56.8) | 221 (43.2) | 512 (100) |

## Main variables descriptives and reliability

The study's main variables, each representing different dimensions, were measured for mean values, standard deviations, standardized mean values, minimum, maximum, and reliability indices. Table 2. For the Medical Artificial Intelligence Readiness Scale (MAIRS), the overall MAIRS, combining all dimensions, had a mean score of 70.53 (SD = 21.32). Participants' MAIRS cognitive dimension averaged 25.11 (SD = 7.75). The MAIRS Ability dimension had a mean score of 25.80 (SD = 8.42). The MAIRS Vision and MAIRS Ethics dimensions had mean scores of 9.25 and 10.36, with standard deviations of 3.35 and 3.30, respectively. The standardized mean showed that the highest score was for the MAIRS ethics domain and the lowest was for the MAIRS vision domain.

The Readiness for Interprofessional Learning Scale (RIPLS) assessed participants' attitudes toward three components of IPC. Professional identity scored a mean of 11.90 (SD = 2.93). The IPC components of T&C and R&R had mean scores and standard deviations of 35.18 (SD = 9.06) and 9.83 (SD = 2.71), respectively. The standardized mean ranked the RIPLS PI the highest and the RIPLS R&R the lowest. The internal reliability assessment was performed using Cronbach's alpha and resulted in moderate to excellent reliability for MAIRS and RIPLS variables. To ensure the reliability of the 'RIPLS R&R' variable, item-total correlation analysis was also performed, and the results were 0.72–0.83, demonstrating good reliability.

## Correlations between professional identity and AI and IPC components

Spearman's rank correlation coefficient analysis was performed to test the relationships between professional identity and AI total and dimensions scores, in addition to IPC components (T&C and R&R). Presented in Table 3. The correlation significance was set at $p < 0.05$ for significance. To interpret the strength of the relationships, Dancey and Reidy's guideline was used in which correlation coefficients within the ranges 0.1–0.3, 0.31–0.6, and greater than 0.6 are considered weak, moderate, and strong, respectively [26].

**Table 2. Main variables descriptives and reliability scores (N = 512).**

| Variable | Min. | Max. | Mean | SD | Standardized mean | Level | Cronbach's alpha |
|---|---|---|---|---|---|---|---|
| MAIRS Cognitive | 8 | 40 | 25.11 | 7.75 | 3.14 | Moderate | 0.93 |
| MAIRS Ability | 8 | 40 | 25.80 | 8.42 | 3.23 | Moderate | 0.96 |
| MAIRS Vision | 3 | 15 | 9.25 | 3.35 | 3.08 | Moderate | 0.93 |
| MAIRS Ethics | 3 | 15 | 10.36 | 3.30 | 3.45 | Moderate | 0.91 |
| MAIRS Total | 22 | 110 | 70.53 | 21.32 | 3.21 | Moderate | 0.98 |
| RIPLS T&C | 9 | 45 | 35.18 | 9.06 | 3.91 | High | 0.91 |
| RIPLS PI | 3 | 15 | 11.90 | 2.93 | 3.97 | High | 0.89 |
| RIPLS R&R | 3 | 15 | 9.83 | 2.71 | 3.28 | Moderate | 0.65 |

**Table 3. Correlations between professional identity and AI and IPC.**

| | Spearman's rank correlation coefficient (ρ) | p-value | Relationship strength |
|---|---|---|---|
| MAIRS Cognitive | .32 | <.01 | Moderate |
| MAIRS Ability | .41 | <.01 | Moderate |
| MAIRS Vision | .38 | <.01 | Moderate |
| MAIRS Ethics | .47 | <.01 | Moderate |
| MAIRS Total | .40 | <.01 | Moderate |
| RIPLS T&C | .79 | <.01 | Strong |
| RIPLS R&R | .37 | <.01 | Moderate |

Accordingly, professional identity demonstrated significantly moderate positive correlations with MAIRS total and subscale scores. This means that as participants' AI readiness increases, their professional identity increases. For IPC components, professional identity was strongly and positively correlated with interprofessional teamwork and collaboration component (r = .79, p < 0.01), and moderately positively correlated with the roles and responsibilities component of interprofessional collaboration.

### Professional identity comparative analyses according to participants' characteristics

The comparative analyses of professional identity according to participants' characteristics were conducted using the Mann-Whitney U test for gender, education, and experience, and the Kruskal-Wallis test for job title (Table 4). Professional identity varied significantly according to gender and experience. More specifically, professional identity was significantly higher among male participants where their mean score was 16.53 compared to female participants who had a mean score of 15.22 (p < 0.05). Participants with experience of 5 years or higher demonstrated significantly higher professional identity. On the other hand, variations in professional identity scores did not significantly vary among different educational levels or professional groups (p > 0.05).

## Discussion

AI and IPC are the gateway of healthcare professionals to the industry and innovation world as well as the excellence of modern healthcare quality [27]. This study is the first of its kind in the region to address such crucial concepts and their relationships with professional identity. To obtain a generalizable sample of the workforce in Jordanian hospitals, this study included participants from a variety of specialties and educational backgrounds. The participants' characteristics showed that about half of the participants were nurses and midwives whereas 23.4% were doctors, 5.5% were pharmacists, and other professionals accounted for 13.7%. These percentages are relatively consistent with the Jordanian health workforce statistics reported by the World Health Organization [28] in which the percentages were 48.5% for nurses and midwives, 29.1% for doctors, 16.3 for pharmacists, and 24.6% for other professionals. These comparisons show that this study demonstrates a sound representation of the Jordanian healthcare workforce.

**Table 4. Nonparametric independent samples test analyses for professional identity.**

| Variable | Category | Min | Max | Mean | SD | Mann-Whitney U | |
|---|---|---|---|---|---|---|---|
| | | | | | | Mean Rank | p-value |
| Gender | Male | 8 | 20 | 16.53 | 3.09 | 281.46 | .003 |
| | Female | 4 | 20 | 15.22 | 4.03 | 241.27 | |
| Education | Diploma or BSc | 4 | 20 | 15.75 | 3.77 | 258.21 | .37 |
| | Postgraduate | 8 | 20 | 15.37 | 3.61 | 236.88 | |
| Experience | <5 years | 4 | 20 | 15.25 | 3.95 | 240.63 | .005 |
| | ≥5 years | 6 | 20 | 16.33 | 3.39 | 277.40 | |

| | | | | | | Kruskal-Wallis | |
|---|---|---|---|---|---|---|---|
| Job Title | Nurse | 4 | 20 | 16.11 | 3.17 | 266.31 | .118 |
| | Physician | 8 | 20 | 14.90 | 4.24 | 232.00 | |
| | Midwife | 4 | 20 | 15.22 | 4.65 | 249.94 | |
| | Pharmacist | 10 | 20 | 16.86 | 3.71 | 299.86 | |
| | Allied Medical Personnel | 12 | 20 | 16.79 | 3.18 | 289.77 | |
| | Computer & IT Specialist | 12 | 19 | 14.88 | 3.32 | 207.06 | |
| | Others | 12 | 20 | 16.07 | 2.59 | 260.43 | |

The examination of professional identity among professionals in healthcare provides crucial insights into the multi-dimensional nature of their protagonists and perceptions within the healthcare setting. Yet, this area is understudied in healthcare literature [29]. Professional identity was high among the participants in this study. The notable high mean scores, spanning various specialties, indicate a comprehensive and well-rounded professional identity among participants. This result is consistent with a previous study that found allied health professionals to have a high professional identity [30]. The study also highlighted that professional identity contributes to organizational success while it is dynamic in the workplace requiring attention to the possible interacting variables.

The participants in this study reported strong interprofessional teamwork and collaboration, and moderate perceived IPC roles and responsibilities. Interestingly. The study also revealed that professional identity is proportionally correlated with IPC components. The significant relationship between professional identity and collaborative teamwork practices as well as roles and responsibilities provide a nuanced understanding of how healthcare practitioners perceive and engage with their professional identities. This result is consistent with van den Broek et al.'s [31] study which showed that professional identity formation contributes positively to interprofessional teamwork and openness to feedback across professionals.

In terms of AI readiness, participants reported moderate readiness in the cognitive, ability, vision, and ethics domains with the ethics domain scoring the highest whereas the vision domain was the lowest. This result is interesting but not surprising because healthcare professionals tend to practice ethical principles and applications in everyday practice, therefore, they have the prerequisites for AI ethical readiness. On the other hand, vision represents foreseeing strengths, limitations, and opportunities, which requires immense knowledge and application of AI tools to enable prospective vision. A review study, conducted on AI readiness, revealed that successful adaptation of AI technologies relies on knowing and understanding, using, evaluating, and considering its ethical concerns [32]. The study also highlighted that AI applications in healthcare require effective training and workplace applications such as using healthcare-AI technologies for disease prevention, diagnosis, treatment, and rehabilitation. Since the participants of the current study demonstrated acceptable readiness for AI, strategies should be tactically planned to improve professionals' competencies and skills to optimize AI usefulness in healthcare settings.

Professional identity was positively correlated with IPC components such as teamwork and collaboration. This result is consistent with a previous study which indicated that professionals with higher professional identity had a broad view of their interprofessional team and valued interprofessional feedback [31]. In addition, the results of the current study revealed a positive relationship between professional identity and AI readiness; indicating that as participants' AI readiness increases, their professional identity rises.

The intersection of professional identity and AI in the healthcare sector sparks a dynamic discourse on the evolving roles of healthcare professionals. To compare the findings of the current study with previous literature, it is paramount to present the discourse in the literature in this area. For example, Char et al. [12] noted that as AI applications increase in healthcare, clinicians' roles decrease which could give AI tools unintended power and affect their professional identity. So, resistance to AI integration may lead to professional identity crises, hindering the full potential of AI to improve patient outcomes [33]. On the other hand, health professionals who embrace the supportive role of AI demonstrate a positive professional identity and are better positioned to manage the evolving dynamics [27]. Therefore, embracing AI as a supportive tool strengthens professional identity, while resistance may lead to identity challenges, highlighting the need for a balanced approach to AI integration in healthcare.

In addition, ethical considerations in the use of AI technologies demand healthcare professionals to make complex decisions regarding patient privacy, data security, and the potential biases rooted in algorithms [12]. Managing these ethical challenges becomes an integral facet of the evolving professional identity within the modern healthcare systems which was evidenced by our result of a positive correlation between professional identity and the ethical dimension of AI readiness.

The study also examined variations in professional identity across different demographic categories. Regarding gender, a significant difference emerged with males demonstrating a higher mean rank compared to females. The observed gender-based disparities in professional identity align with a recent study among nurses [34]. In addition, a significant difference in professional identity scores was found favoring the higher experiential level. This result is consistent with Philippa et al.'s [35] study highlighting that professional development and time spent working in the profession were positive forces contributing to the way practitioners perceive their professions within the work context. Moreover, Wyatt et al [36] suggested using modern technologies and simulation to mimic real-life experiences to improve the professional identity of healthcare professionals.

## Strengths and limitations of the study

This study is the first of its kind to investigate professional identity in the modern healthcare context. The study was conducted across professions contributing to the interprofessional healthcare literature. However, this study has some limitations that should be addressed. The current study used convenience sampling which inherently poses bias because professionals with limited knowledge on the topic may feel reluctant to participate in the study. Also, this study used a cross-sectional design. Future studies may use longitudinal research design to capture how professional identity evolves with time, experience, and advancements in healthcare contexts.

## Conclusions and implications

The study sets the grounding roles to develop the healthcare workforce's professional identity within the dynamic healthcare environment in the age of AI and IPC. The study revealed that healthcare professionals reported high professional identity and teamwork collaborative practices, and moderate AI readiness. This highlights areas of development for healthcare managers and practitioners. Using AI Interprofessional collaboration-based training may be ideal, targeting both AI domains and IPC components while preserving a positive professional identity. The study also emphasized the positive relationship between professional identity and AI, IPC, and experience.

Health-related educators can benefit from the study results in curriculum development to utilize educational opportunities that align with the broader goals of enhancing patient care and fostering cohesive teamwork within the healthcare industry. Future longitudinal and interventional studies will enhance the evaluation of the progress of these paramount concepts among healthcare professionals.

AcknowledgmentThe authors are thankful to the Deanship of Research at Jordan University of Science and Technology for their support.

## Author contributions

**Conceptualization:** Wafa'a Ta'an, Sadeq Damrah, Mohammed M. Al-Hammouri, Brett Williams.

**Data curation:** Wafa'a Ta'an, Sadeq Damrah.

**Formal analysis:** Wafa'a Ta'an.

**Funding acquisition:** Wafa'a Ta'an.

**Methodology:** Wafa'a Ta'an, Sadeq Damrah, Mohammed M. Al-Hammouri, Brett Williams.

**Project administration:** Wafa'a Ta'an.

**Visualization:** Wafa'a Ta'an.

**Writing – original draft:** Wafa'a Ta'an, Sadeq Damrah, Mohammed M. Al-Hammouri, Brett Williams.

**Writing – review & editing:** Wafa'a Ta'an, Sadeq Damrah, Mohammed M. Al-Hammouri, Brett Williams.

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
