## [Editor Report · Decision Letter 0]

25 Nov 2024

PONE-D-24-39174Professional Identity in the Age of AI and Interprofessional Collaboration among Healthcare PractitionersPLOS ONE

Dear Dr. Ta'an,

Thank you for submitting your manuscript to PLOS ONE. After careful consideration, we feel that it has merit but does not fully meet PLOS ONE’s publication criteria as it currently stands. Therefore, we invite you to submit a revised version of the manuscript that addresses the points raised during the review process.

**ACADEMIC EDITOR: **

After an initial evaluation, I commend your efforts to address the timely and relevant topic of professional identity in the evolving landscape of healthcare with the integration of artificial intelligence and interprofessional collaboration.

However, I have identified several areas that require significant improvement before the manuscript can be sent for peer review. Please find my detailed comments below:

Flow and Readability: The manuscript seems to lack a coherent flow, making it challenging to follow the narrative. This could hinder reviewers' ability to assess the study effectively. Suggest restructuring the content to enhance clarity and logical progression.

Methodological Concerns: The reliability of the tools used, particularly the Medical Artificial Intelligence Readiness Scale (MAIRS) and the Readiness for Interprofessional Learning Scale (RIPLS), raises issues. For instance, some subscales have low Cronbach’s alpha values, which suggest poor internal consistency and may affect the credibility of the findings. Please justify the use of tools with low reliability scores or consider alternative validated instruments.

Data Availability: The data availability statement does not meet the PLOS ONE guidelines, as it should clearly indicate unrestricted access to data, barring exceptional cases. Ensure the data availability aligns with PLOS ONE’s policies, providing clear access to data or a valid justification for any restrictions.

Specify Data Collection Period: The manuscript does not mention when the data was collected. This information is essential for understanding the study's context and potential relevance to the current state of the field.

We look forward to receiving your revised manuscript.

Kind regards,

Mohd Ismail Ibrahim, MCom.Med

Academic Editor

PLOS ONE

2. Thank you for stating the following financial disclosure: [This study is funded by the Deanship of Research at Jordan University of Science and Technology (Research Grant ID: 20230635).]. Please state what role the funders took in the study. If the funders had no role, please state: "The funders had no role in study design, data collection and analysis, decision to publish, or preparation of the manuscript." If this statement is not correct you must amend it as needed. Please include this amended Role of Funder statement in your cover letter; we will change the online submission form on your behalf.

3. In the online submission form, you indicated that [The data underlying the results presented in the study are available from the principal investigator upon reasonable request.]. All PLOS journals now require all data underlying the findings described in their manuscript to be freely available to other researchers, either 1. In a public repository, 2. Within the manuscript itself, or 3. Uploaded as supplementary information. This policy applies to all data except where public deposition would breach compliance with the protocol approved by your research ethics board. If your data cannot be made publicly available for ethical or legal reasons (e.g., public availability would compromise patient privacy), please explain your reasons on resubmission and your exemption request will be escalated for approval.
---

## [Author Response · Author response to Decision Letter 0]

10 Dec 2024

Response to editor and reviewers is attached in the submission. Thank you.

---

## [Decision Letter · Decision Letter 1]

26 Dec 2024

PONE-D-24-39174R1Professional Identity in the Age of AI and Interprofessional Collaboration among Healthcare PractitionersPLOS ONE

Dear Dr. Ta'an,

Thank you for submitting your manuscript to PLOS ONE. After careful consideration, we feel that it has merit but does not fully meet PLOS ONE’s publication criteria as it currently stands. Therefore, we invite you to submit a revised version of the manuscript that addresses the points raised during the review process.

We look forward to receiving your revised manuscript.

Kind regards,

Mohd Ismail Ibrahim, MCom.Med

Academic Editor

PLOS ONE

Reviewers' comments:

Reviewer's Responses to Questions

**Comments to the Author**

1. If the authors have adequately addressed your comments raised in a previous round of review and you feel that this manuscript is now acceptable for publication, you may indicate that here to bypass the “Comments to the Author” section, enter your conflict of interest statement in the “Confidential to Editor” section, and submit your "Accept" recommendation.

Reviewer #1: All comments have been addressed

Reviewer #2: (No Response)

2. Is the manuscript technically sound, and do the data support the conclusions?

Reviewer #1: Yes

Reviewer #2: Partly

3. Has the statistical analysis been performed appropriately and rigorously? 

Reviewer #1: Yes

Reviewer #2: No

4. Have the authors made all data underlying the findings in their manuscript fully available?

Reviewer #1: Yes

Reviewer #2: Yes

5. Is the manuscript presented in an intelligible fashion and written in standard English?

Reviewer #1: Yes

Reviewer #2: Yes

6. Review Comments to the Author

Reviewer #1: Dear Authors.

This is a very well written manuscript.

We recommend to compare your result to results by previous researchers.

Detail in spreadsheet.

Reviewer #2: 1. The goal of this manuscript was to examine professional identity and its relationships with AI readiness domains and interprofessional collaboration components, however, the title “Professional Identity in the Age of AI and Interprofessional Collaboration among Healthcare Practitioners”, does not clearly address this point. It is recommended to revise the title.

2. The manuscript does not mention literature evidence on the relationships between professional identity and AI readiness. This could be added in the background.

3. In RIPLS, items 10-12 are Negative Professional Identity and 13-16 are Positive Professional Identity�how to get “Professional identity scored a mean of 15.72 (SD = 3.75)” in line 245�Please provide more context or explain in detail.

4. line 210-216, Table 3. The preliminary analysis revealed that the professional identity scores were not normally distributed among the participants nor among the groups of participants which require nonparametric comparative analyses, why used Pearson r correlation coefficient to examine the relationships between professional identity and other variables (AI readiness and IPC components)? Why not the Spearman's rank correlation coefficient?

5. Line 166�”This questionnaire includes 17 items”, but according to reference 19, the Readiness for Interprofessional Learning Scale (RIPLS) includes 19 items. Please review carefully.

6. Line 167�”which are team teamwork and collaboration (T&C)”, it seems like the term “team” could be removed.

7. The "ai" in the line 252 should be capitalized.

7. PLOS authors have the option to publish the peer review history of their article (what does this mean? ). If published, this will include your full peer review and any attached files.

**Do you want your identity to be public for this peer review?** For information about this choice, including consent withdrawal, please see our Privacy Policy .

Reviewer #1: **Yes: ** Renan Prasta Jenie

Reviewer #2: No

---

## [Author Response · Author response to Decision Letter 1]

29 Dec 2024

Provided as an attachment. Thank you!

---

## [Decision Letter · Decision Letter 2]

12 Jan 2025

PONE-D-24-39174R2Professional Identity and its Relationships with AI Readiness and Interprofessional CollaborationPLOS ONE

Dear Dr. Ta'an,

Thank you for submitting your manuscript to PLOS ONE. After careful consideration, we feel that it has merit but does not fully meet PLOS ONE’s publication criteria as it currently stands. Therefore, we invite you to submit a revised version of the manuscript that addresses the points raised during the review process.

We look forward to receiving your revised manuscript.

Kind regards,

Mohd Ismail Ibrahim, MCom.Med

Academic Editor

PLOS ONE

Journal Requirements:

Reviewers' comments:

Reviewer's Responses to Questions

**Comments to the Author**

1. If the authors have adequately addressed your comments raised in a previous round of review and you feel that this manuscript is now acceptable for publication, you may indicate that here to bypass the “Comments to the Author” section, enter your conflict of interest statement in the “Confidential to Editor” section, and submit your "Accept" recommendation.

Reviewer #1: (No Response)

Reviewer #2: All comments have been addressed

2. Is the manuscript technically sound, and do the data support the conclusions?

Reviewer #1: Yes

Reviewer #2: Yes

3. Has the statistical analysis been performed appropriately and rigorously? 

Reviewer #1: Yes

Reviewer #2: Yes

4. Have the authors made all data underlying the findings in their manuscript fully available?

Reviewer #1: Yes

Reviewer #2: Yes

5. Is the manuscript presented in an intelligible fashion and written in standard English?

Reviewer #1: Yes

Reviewer #2: Yes

6. Review Comments to the Author

Reviewer #1: Thanks for abiding my previous review.

As previous review, this article already well written, sans some detail which I have put in spreadsheet.

We hope for speedy publication

Reviewer #2: (No Response)

7. PLOS authors have the option to publish the peer review history of their article (what does this mean? ). If published, this will include your full peer review and any attached files.

**Do you want your identity to be public for this peer review?** For information about this choice, including consent withdrawal, please see our Privacy Policy .

Reviewer #1: **Yes: ** Renan Prasta Jenie

Reviewer #2: **Yes: ** Guo Huang

---

## [Author Response · Author response to Decision Letter 2]

21 Feb 2025

Thank you for providing valuable feedback to improve the quality of our manuscript. We have revised the references list and updated the link for the inactive document’s link.

The Previous reference:

[28] World Health Organization (2017). Health workforce snapshot: Jordan. Retrieved January 20, 2024 from: https://rho.emro.who.int/sites/default/files/Profiles-briefs-files/Jordan-HWF-Snapshot_2020.pdf

Was updated as below:

[28] World Health Organization (2017). Health workforce snapshot: Jordan. Retrieved January 20, 2024 from: https://applications.emro.who.int/docs/WHOEMHRH642E-eng.pdf

Kind Regards,

Wafa'a Ta'an

---

## [Editor Report · Decision Letter 3]

25 Feb 2025

PONE-D-24-39174R3Professional Identity and its Relationships with AI Readiness and Interprofessional CollaborationPLOS ONE

Dear Dr. Ta'an,

Thank you for submitting your manuscript to PLOS ONE. After careful consideration, we feel that it has merit but does not fully meet PLOS ONE’s publication criteria as it currently stands. Therefore, we invite you to submit a revised version of the manuscript that addresses the points raised during the review process.

**ACADEMIC EDITOR: ** Please address the concerns raised by the reviewer. The reviewer noted a few minor issues in the abstract and discussion that require improvement by the authors (please refer to the attachment). Prompt corrections will expedite the publication process.

We look forward to receiving your revised manuscript.

Kind regards,

Mohd Ismail Ibrahim, MCom.Med

Academic Editor

PLOS ONE
---

## [Author Response · Author response to Decision Letter 3]

26 Mar 2025

Responses to Reviewer’s Comments

Dear Editor and Reviewers,

We would like to extend our thanks for giving us a chance to resubmit the revised manuscript and to reviewers for reviewing the manuscript (PONE-D-24-39174R3), entitled “Professional Identity and its Relationships with AI Readiness and Interprofessional Collaboration.” We reviewed the manuscript based on the reviewer’s attached Excel file. We read and understood their comments, and we believe that they are fully addressed. All changes have been made and attached as a ‘Manuscript with Track Changes’.

Reviewer’s Comment Response Reference

Abstract: Contain about 200 to 250 words.

Revised to adhere to the guidelines

Abstract: Please add timeframe. We suggest to add quantitative values

Added to the abstract

Discussions: We suggest to add some comparison with previous researches

Comparisons were added and highlighted

Thank you very much for your comments and efforts to improve the quality of the manuscript.

Wafaa Taan

---

## [Editor Report · Decision Letter 4]

28 Mar 2025

Professional Identity and its Relationships with AI Readiness and Interprofessional Collaboration

PONE-D-24-39174R4

Dear Dr. Ta'an,

We’re pleased to inform you that your manuscript has been judged scientifically suitable for publication and will be formally accepted for publication once it meets all outstanding technical requirements.

Kind regards,

Mohd Ismail Ibrahim, MCom.Med

Academic Editor

PLOS ONE
---

## [Editor Report · Acceptance letter]

PONE-D-24-39174R4

PLOS ONE

Dear Dr. Ta'an,

I'm pleased to inform you that your manuscript has been deemed suitable for publication in PLOS ONE. Congratulations! Your manuscript is now being handed over to our production team.

Kind regards,

on behalf of

Dr. Mohd Ismail Ibrahim

Academic Editor

PLOS ONE